# Ataxin-1 oligomers induce local spread of pathology and decreasing them by passive immunization slows Spinocerebellar ataxia type 1 phenotypes

Cristian A Lasagna-Reeves[1,2], Maxime WC Rousseaux[1,2], Marcos J Guerrero-Munoz[3], Luis Vilanova-Velez[1,2], Jeehye Park[1,2], Lauren See[1,2], Paymaan Jafar-Nejad[1,2], Ronald Richman[1,2,4], Harry T Orr[5], Rakez Kayed[3], Huda Y Zoghbi[1,2,4,6]*

[1]Department of Molecular and Human Genetics, Baylor College of Medicine, Houston, United States; [2]Jan and Dan Duncan Neurological Research Institute, Texas Children's Hospital, Houston, United States; [3]Department of Neurology, University of Texas Medical Branch, Galveston, United States; [4]Howard Hughes Medical Institute, Baylor College of Medicine, Houston, United States; [5]Institute for Translational Neuroscience, University of Minnesota, Minnesota, United States; [6]Department of Neuroscience, Baylor College of Medicine, Houston, United states

**Abstract** Previously, we reported that ATXN1 oligomers are the primary drivers of toxicity in Spinocerebellar ataxia type 1 (SCA1; Lasagna-Reeves et al., 2015). Here we report that polyQ ATXN1 oligomers can propagate locally in vivo in mice predisposed to SCA1 following intracerebral oligomeric tissue inoculation. Our data also show that targeting these oligomers with passive immunotherapy leads to some improvement in motor coordination in SCA1 mice and to a modest increase in their life span. These findings provide evidence that oligomer propagation is regionally limited in SCA1 and that immunotherapy targeting extracellular oligomers can mildly modify disease phenotypes.

*For correspondence: hzoghbi@bcm.edu

## Introduction

We recently demonstrated that soluble oligomers play an important role in Spinocerebellar ataxia type 1 (SCA1), a neurodegenerative disease caused by expansion of a CAG repeat that encodes for glutamine (Q) in ataxin-1 (ATXN1)(*Lasagna-Reeves et al., 2015*). In the same study, we showed that ATXN1 oligomers are internalized into the cell and seed the formation of new ATXN1 oligomers (*Lasagna-Reeves et al., 2015*). This 'seeding' concept has been explored in several other proteinopathies and is believed to be one of the drivers of neurodegeneration (*Frost et al., 2009*; *Munch et al., 2011*; *Holmes and Diamond, 2012*). What's more, evidence has mounted that soluble aggregates of amyloidogenic proteins including Tau, Aβ and α–synuclein, can propagate pathology from cell-to-cell in a prion-like fashion, thereby causing pathology progression from one brain region to the another in a disease-specific pattern (*Hardy and Revesz, 2012*; *Holmes and Diamond, 2012*; *Guo and Lee, 2014*). Passive immunotherapy has been recently proposed as a feasible strategy to inhibit pathology propagation in mouse models for proteinopathy (*Banks et al., 2007*; *Chai et al., 2011*; *Masliah et al., 2011*; *Yanamandra et al., 2013*; *Castillo-Carranza et al., 2014*; *Games et al., 2014*; *Tran et al., 2014*). Despite these advances, it is not clear whether a mostly nuclear protein

like ATXN1 will propagate from cell-to-cell in vivo and whether passive immunotherapy targeting oligomers will modify disease course.

In the present study, we report that polyQ ATXN1 oligomers act as a seed by inducing disease propagation to neighboring but not to distal cells and demonstrate that this cell-to-cell spread can be blocked using passive immunotherapy.

## Results

### ATXN1 oligomers seed the formation of new endogenous ATXN1 oligomers in vivo

We recently demonstrated that in $Atxn1^{154Q/+}$ mice ATXN1 oligomers are restricted to focal sub-populations of Purkinje cells (PCs) and are not evenly present throughout the cerebellum. Notably, this focal distribution coincided with cellular toxicity (*Lasagna-Reeves et al., 2015*). This observation, together with the finding that ATXN1 oligomers are able to penetrate cells in culture and seed the formation of new ATXN1 oligomers led us to hypothesize that if ATXN1 oligomers propagate in vivo, intracerebral injection of brain extract from a symptomatic SCA1 mouse into a disease-free mouse would predispose the latter to develop neuropathology. To test this hypothesis, we injected cerebellar extract (10 µg, 2.5 µL) either from $Atxn1^{154Q/+}$ or wild-type mice into the deep cerebellar nuclei of wild type, $Atxn1^{-/-}$ and $Atxn1^{78Q/+}$ mice (*Figure 1A*). $Atxn1^{78Q/+}$ mice express one allele of murine Atxn1 with a 78Q expansion. These mice do not display behavioral abnormalities nor any neuropathology, including ATXN1 inclusions, indicating that a single copy of 78Q-Atxn1 is insufficient to generate disease within the short lifespan of a mouse (*Lorenzetti et al., 2000*). These characteristics make this mouse an ideal model to determine the seeding abilities of ATXN1 soluble oligomers from $Atxn1^{154Q/+}$ mice lysates.

Before performing injections, we confirmed by western blot analysis that the injected material from $Atxn1^{154Q/+}$ mouse cerebella was indeed enriched in oligomers (*Figure 1A*). Nine different groups of mice (each constituting 8 animals) were injected at 3 months of age (*Figure 1B*). Three months after injection, ELISA revealed that cerebellar extracts from the SCA1 $Atxn1^{154Q/+}$ mice caused a two-to-three fold increase in oligomer formation in the cerebellum and brain stem of $Atxn1^{78Q/+}$ mice compared to other groups (*Figure 1C*, left panels). Oligomer levels in the cortex, which were lower to begin with than in the cerebellum and brainstem, did not significantly change in any of the groups following injection of cerebellar lysate. Neuropathology confirmed that oligomers were located in cerebellum and brain stem, but not in cortex, hippocampus, striatum or olfactory bulb (data not shown). ATXN1 oligomers thus appear to propagate only to neighboring areas, and, at least under these conditions, trans-synaptic propagation was not detected. Nevertheless, we can't exclude the possibility that a higher concentration of inoculum and/or a longer incubation time might lead to oligomer propagation to farther areas. In all of the groups of mice tested, we observed similar amounts of total ATXN1 (*Figure 1C*, right panels), which indicates that the increase of oligomers in the cerebellum and brain stem of $Atxn1^{78Q/+}$ mice was produced by a seeding effect of ATXN1-154Q soluble oligomers rather than by inducing more expression of ATXN1-78Q. To evaluate whether or not pathology propagation was triggered by oligomers presented in the injected material from $Atxn1^{154Q/+}$ we performed a cell-based seeding assay using the injected materials as previously described (*Lasagna-Reeves et al., 2015*). Specifically, we added the injected material from $Atxn1^{154Q/+}$ alone or pre-incubated with the anti-oligomer antibody F11G3 to cells that express ATXN1(82Q) fused to mRFP. After a 10 hr incubation, we then quantified the amount of oligomeric inclusions. In comparison with cells incubated with the injected material from wild type or non-treated group, the cells exposed only to the injected material from $Atxn1^{154Q/+}$ developed the largest proportion of oligomeric inclusions (37.25%). Moreover, when cells were exposed to the injected material from $Atxn1^{154Q/+}$ pre-incubated with F11G3, the proportion of oligomeric inclusions (15.25%) did not increase in comparison with the control groups thus suggesting that the active material in the injected lysate are the oligomers (*Figure 1—figure supplement 1*). In addition, this oligomeric pathology did not result from the surgical procedure itself: only PCs of $Atxn1^{78Q/+}$ knockin mice injected with cerebellar extract from $Atxn1^{154Q/+}$ mice, not those injected with WT cerebellar extract, showed an increase in oligomers (*Figure 1D*). The observation that no oligomers were detected in PCs from either wild-type or $Atxn1$ null mice injected with $Atxn1^{154Q/+}$ lysates suggests

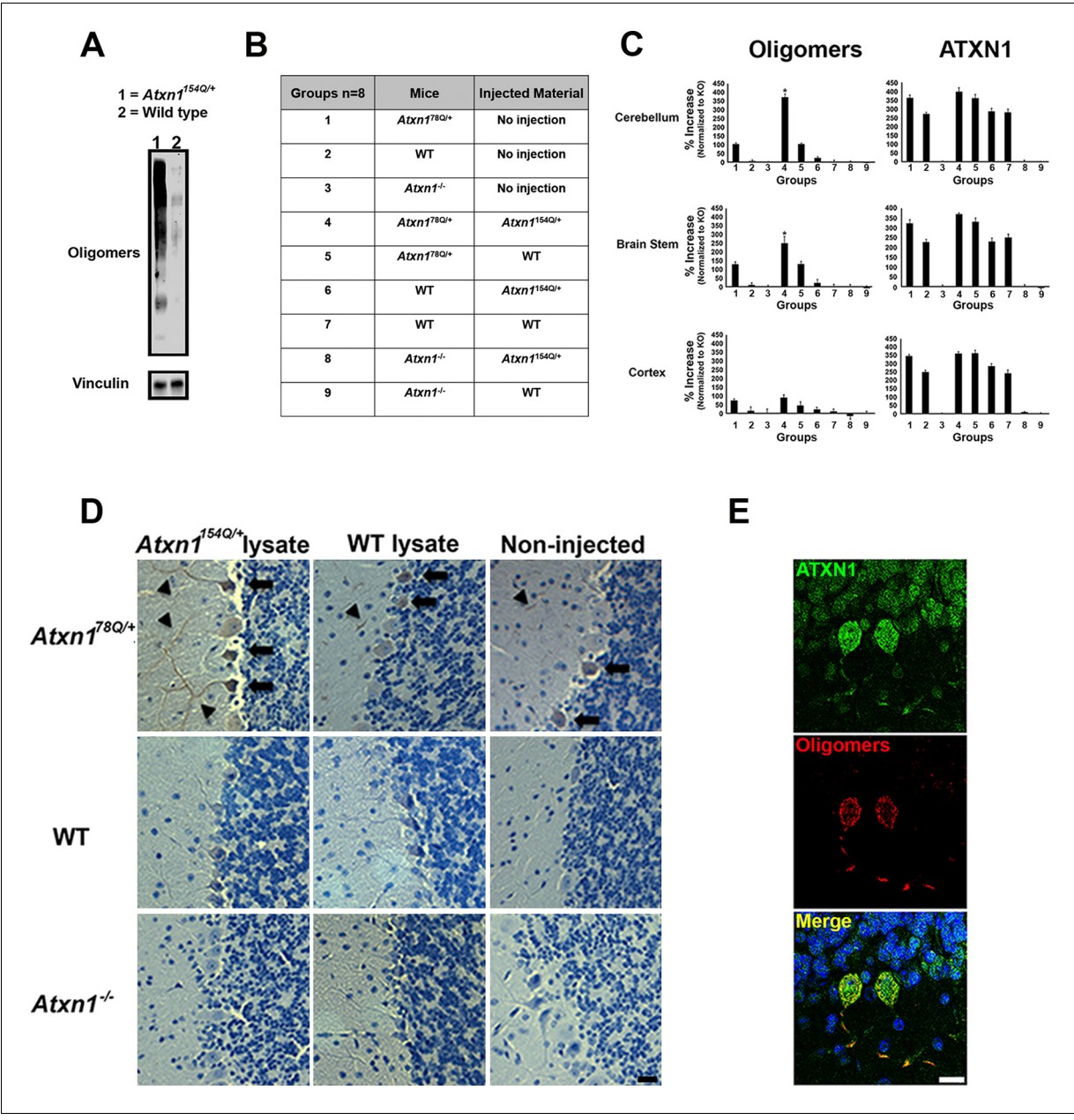

**Figure 1.** ATXN1 oligomers propagate further ATXN1 oligomerization in vivo. (**A**) Representative western blot brain lysates used for in vivo injections. F11G3 was used to detect oligomers (WT and *Atxn1*$^{154Q/+}$, cerebellar samples). (**B**) Table of mouse genotypes and treatments used in the in vivo propagation assay. (**C**) ELISA for oligomers (F11G3, left panels) and ATXN1 (11750, right panels) was performed on WT, *Atxn1*$^{78Q/+}$ and *Atxn1*$^{-/-}$ mice injected with cerebellar lysate (WT or *Atxn1*$^{154Q/+}$) in the indicated brain regions. *x* axis indicated groups from (**B**) * denotes $p<0.05$, ANOVA followed by Bonferroni's *post hoc* test. (**D**) Representative histological staining for oligomers (F11G3) in groups indicated in (**B**) in the cerebellum. Arrowheads indicate the accumulation of oligomers in dendrites, arrows indicate their presence in the soma of PCs. Scale bar 15 μm. (**E**) Double staining using anti-ATXN1 antibody (green) and anti-oligomer antibody (red) confirmed the presence of ATXN1 oligomers in Purkinje cells of *Atxn1*$^{78Q/+}$ mice injected with *Atxn1*$^{154Q/+}$ cerebellar soluble fraction. Scale bar 15 μm.

The following figure supplements are available for figure 1:

**Figure supplement 1.** Anti-oligomer antibody F11G3 blocks the formation of oligomers in cells treated with ATXN1 oligomers from Atxn1154Q/+mouse cerebellum.

**Figure supplement 2.** Oligomer propagation is accompanied by slight motor deficit in Atxn178Q/+ mice.

that host expression of polyQ-expanded ATXN1 is necessary for the de novo formation of ATXN1 oligomers in PCs. Immunofluorescence confirmed that the oligomers detected in $Atxn1^{78Q/+}$PCs were indeed ATXN1 oligomers (*Figure 1E*). Overall, these data suggest that the ATXN1 oligomers present in the injected material from $Atxn1^{154Q/+}$ are responsible for inducing propagation of ATXN1 oligomers in $Atxn1^{78Q/+}$mice. Nevertheless, it is possible that other factors present in the inoculum play an important role in promoting propagation. Further, downstream exploration of the precise biochemical composition of these lysates will surely shed insight into this matter.

To determine if oligomer propagation was accompanied by motor deficit, we performed the rotarod assay in mice harboring brain lysate for 3 months. Even though the performance varied during training, there was no evidence of sustained deficit in the $Atxn1^{78Q/+}$knockin mice injected with cerebellar extract from $Atxn1^{154Q/+}$ mice (*Figure 1—figure supplement 2*). Despite the fact that oligomers propagate through the cerebellum in $Atxn1^{78Q/+}$knockin mice, the newly formed 78Q ATXN1 oligomers appear insufficient to induce motor deficit or degeneration (data not shown). This lack of toxicity could be because in $Atxn1^{78Q/+}$knockin adult mice, the neurons are relatively healthy and thus can counteract the toxic effect of newly formed oligomers. In contrast, in $Atxn1^{154Q/+}$mice, the neurons become dysfunctional by four weeks (*Lasagna-Reeves et al., 2015*) rendering them more susceptible to the toxicity of oligomers. Another possibility is that the newly formed 78Q ATXN1 oligomers adopt a different structure or conformation that is less toxic than the 154Q ATXN1 oligomers. Further studies will be necessary to determine if these newly formed ATXN1 oligomers are solely derived from 78Q ATXN1 molecules or whether they contain some persistent 154Q ATXN1 seeds.

## Passive immunotherapy decreases ATXN1 oligomer pathology and improves motor coordination

Given the ability of the anti-oligomer antibody F11G3 to inhibit the internalization of ATXN1 oligomer complexes from $Atxn1^{154Q/+}$ cerebellum in vitro (*Lasagna-Reeves et al., 2015*), we postulated that anti-oligomer (F11G3) immunotherapy could reduce the load of ATXN1 oligomers and mitigate the motor impairment observed in the $Atxn1^{154Q/+}$ model. Because $Atxn1^{154Q/+}$ mice develop motor incoordination as early as 5 weeks of age (*Watase et al., 2002*), we began treatment at 4 weeks of age, where $Atxn1^{154Q/+}$ mice already have a modest accumulation of oligomers in the cerebellum (*Figure 2—figure supplement 1*). Wild-type and $Atxn1^{154Q/+}$ mice were injected intraperitoneally with F11G3 or control IgM antibodies (5 mg/Kg) once a week for 6 weeks. One week after the last injection, we performed the rotarod assay, sacrificed the mice and performed pathological and biochemical analyses. Injected mice were separated into two cohorts, one for biochemical and pathological analysis and one for behavioral and survival analysis. We focused our pathological examination on the cerebellum. Brain sections were immunostained with the anti-oligomer antibody, A-11 (*Kayed et al., 2003*), to ensure depletion of oligomers at the site of interest. $Atxn1^{154Q/+}$ mice treated with the anti-oligomer antibody showed fewer PCs with oligomers than the control group (*Figure 2A and B*). We further analyzed the cerebella by ELISA and noted a decrease in the amount of oligomers and ATXN1 (*Figure 2C and D*). Notably, the immunotherapy did not affect the formation and total number of nuclear inclusions in the cortex (*Figure 2E*): either these cortical inclusions are not preceded by oligomers or the antibody did not get internalized into these cells in vivo. Considering our previous study where we demonstrated that the anti-oligomer antibody blocks the seeding effect of ATXN1 oligomers in cell culture without getting internalized into the cell (*Lasagna-Reeves et al., 2015*), we suggest that the anti-oligomer antibody arrests the propagation of ATXN1 oligomer complexes to neighboring areas by targeting extracellular oligomeric entities rather than directly targeting intracellular oligomers. To confirm that the immunotherapy was indeed targeting ATXN1 oligomers, we analyzed the treated and control samples by western blot using F11G3 and anti-ATXN1 antibody (*Figure 3A and B*). The western blot analysis revealed a decrease in ATXN1 oligomers in the treated group, but no observable change at the level of monomeric ATXN1 (*Figure 3A and B*).

To determine the functional consequences of this immunotherapy, we tested motor performance in treated and control $Atxn1^{154Q/+}$ mice. Rotarod analysis revealed a mildly significant improvement in coordination following six weeks of anti-oligomer immunotherapy (*Figure 3C*). Although treated mice still possessed a motor deficit in comparison with wild type mice, continued weekly

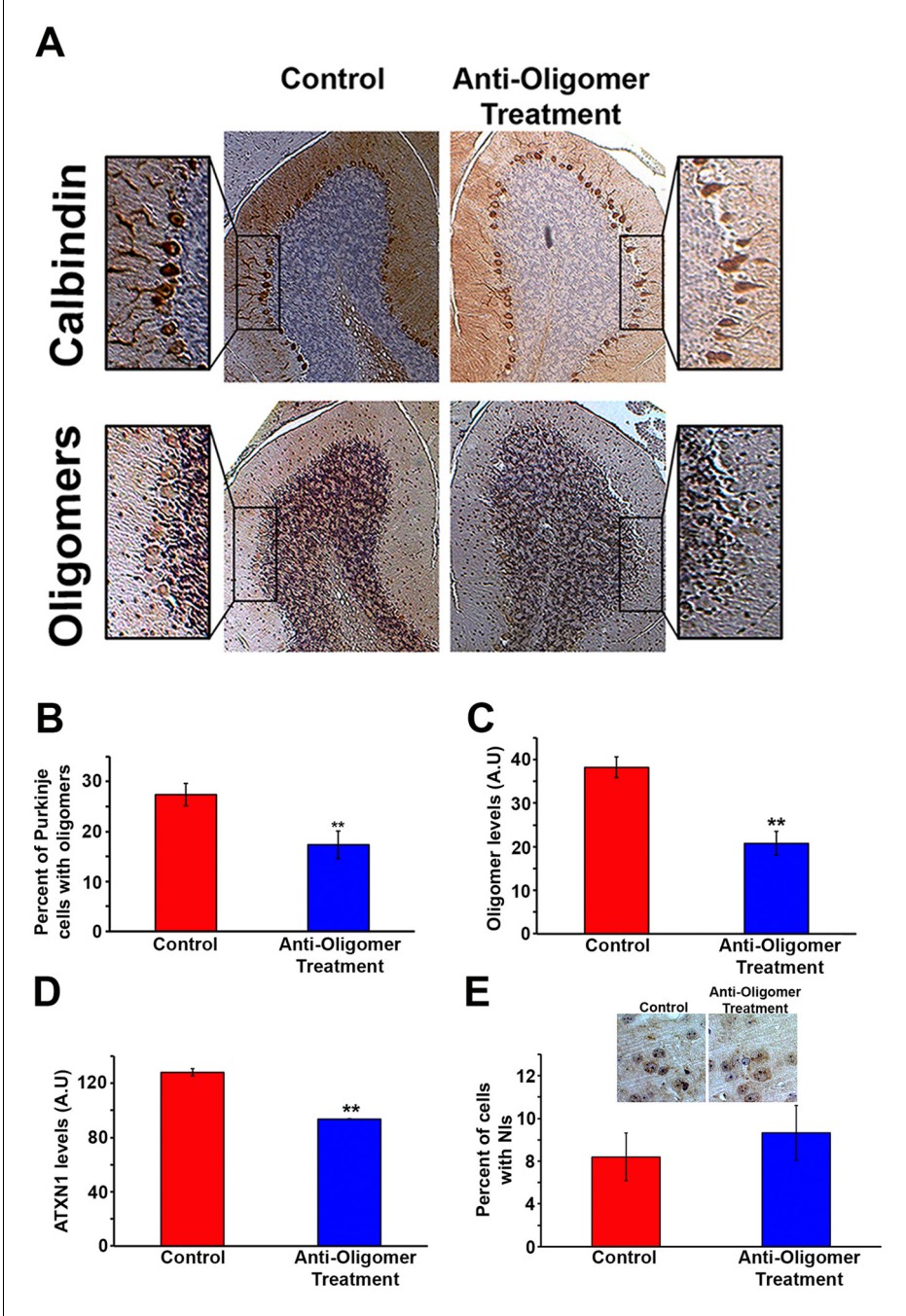

**Figure 2.** Anti-oligomer immunotherapy decreases pathology in vivo. (**A**) Histological staining of PCs (Calbindin, top panels) and Oligomers (A11, bottom panel) of control (IgM) and treated (F11G3) *Atxn1$^{154Q/+}$* mice. Adjacent sections were used for comparison. (**B**) Quantification of (**A**), showing the percentage of PCs with oligomers. Data are represented as mean ± SEM., and ** denotes p<0.01, Student's T-test. (**C**) ELISA for oligomer levels (A11) in the cerebellum of treated mice and the control group. Data are represented as mean ± SEM., and ** denotes p<0.01, Student's T-test. (**D**) ELISA for ATXN1 (11750) levels in the cerebellum of treated mice and the control group. Data are represented as mean ± SEM., and ** denotes p<0.01, Student's T-test. (**E**) Immunotherapy in the cortex of *Atxn1$^{154Q/+}$* mice produced no significant change in the number of cells with nuclear inclusions (NIs, stained with 11750 antibody).

The following figure supplement is available for figure 2:

**Figure supplement 1.** Oligomers are detected in Atxn1$^{154Q/+}$ cerebellum at four weeks of age.

immunotherapy treatments in *Atxn1*$^{154Q/+}$ mice throughout their lifespan extended survival by approximately 3.5 weeks in comparison with the control-treated *Atxn1*$^{154Q/+}$ mice (*Figure 3D*).

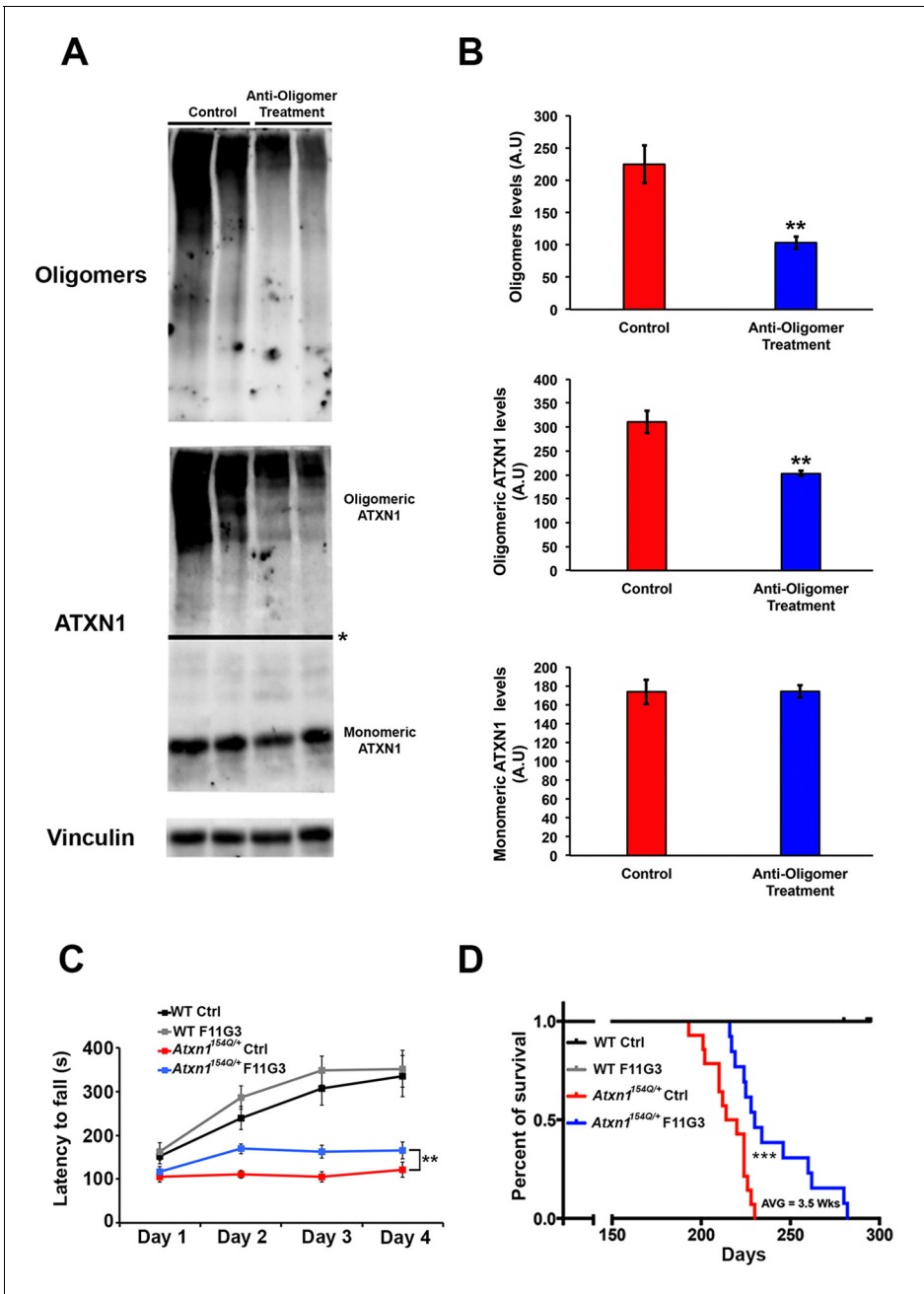

**Figure 3.** Anti-oligomer immunotherapy improves motor deficits and survival in vivo. (A–B) Western blot detecting ATXN1 oligomers (F11G3, top panel) and ATXN1 monomer (11750, middle panel) in the cerebellum of *Atxn1*$^{154Q/+}$ following immunotherapy (anti-oligomer or control). * in (A) indicated change in exposure of the membrane. (B) Quantification of relative levels of oligomeric and monomeric ATXN1 from (A). ** denotes p<0.01, Student's T-test. (C) Rotarod assay in all treatment groups over a four-day period (four trials per day, averaged) 6 weeks following onset of immunotherapy (mice 10 weeks of age). *n* = 12 per genotype; ** denotes p<0.01, ANOVA followed by Tukey's *post hoc* test. (D) Kaplan-Meier survival curve shows that animals treated with anti-oligomer immunotherapy (blue line) lived, on average, 3.5 weeks longer than control animals (red line). No death was observed in WT mice receiving immunotherapy (black and grey lines). *** denotes p<0.001, Log-rank (Mantel-Cox) test. *n* = 12 per genotype.

## Discussion

Our previous study showed that ATXN1 oligomeric complexes were able to penetrate cells in culture and seed the formation of new ATXN1 oligomers (*Lasagna-Reeves et al., 2015*). It is widely accepted that amyloid formation is characterized by such seeding (*Serio et al., 2000*; *Pedersen et al., 2004*). This phenomenon has been explored in cell culture assays, several of which have demonstrated that extracellular aggregates that are internalized into the cell induce aggregation of intracellular proteins (*Frost et al., 2009*; *Munch et al., 2011*; *Holmes and Diamond, 2012*). Moreover, the concept of cell-to-cell spread of many amyloidogenic proteins in vivo has been demonstrated by the progression of protein aggregate pathology from one brain region to another in a disease-specific pattern (*Hardy and Revesz, 2012*; *Holmes and Diamond, 2012*). In our study, however, we found that injected oligomers induced formation of new ATXN1 oligomers only in areas proximal to the injection site. It thus seems unlikely that ATXN1 oligomers propagate transsynaptically in SCA1, as has been suggested with other neurodegenerative diseases (*Harris et al., 2010*; *de Calignon et al., 2012*; *Liu et al., 2012*; *Pecho-Vrieseling et al., 2014*). Rather, we suggest ATXN1 propagates through a secretion and reuptake mechanism between neighboring cells as has been proposed for other proteinopathies (*Guo and Lee, 2014*). Furthermore, no obvious signs of degeneration or motor incoordination were observed in the injected mice. This suggests that the cell must continuously express the toxic entity to trigger degeneration and behavioral deficits in mice. This notion is supported by the observation that immunotherapy targeting the ATXN1 oligomers slightly modified the phenotype in the $Atxn1^{154Q/+}$ mice: the antibody hindered the local propagation of ATXN1 oligomers. This treatment is not curative, however, because neurons expressing polyQ ATXN1 will still form their own toxic oligomeric entities, and some non-oligomeric forms of PolyQ ATXN1 might contribute to toxicity.

Passive immunotherapy has been proposed as a feasible strategy to inhibit pathogenesis in mouse models for AD and PD (*Banks et al., 2007*; *Chai et al., 2011*; *Masliah et al., 2011*; *Yanamandra et al., 2013*; *Castillo-Carranza et al., 2014*; *Games et al., 2014*). Our findings suggest that such therapy might slightly impact disease progression by blocking the propagation of these oligomeric entities, but to halt or reverse symptoms it would likely be necessary to incorporate a joint therapy that targets the root cause of disease, the abnormal accumulation of polyQ ATXN1 (*Park et al., 2013*). Altogether, our findings revealed that ATXN1 oligomer propagation contributes to SCA1 pathogenesis, and that such propagation especially in the vicinity of affected cells, could be targeted through passive immunotherapy. That the benefit from such therapy was small highlights the need to develop therapies that target ATXN1, the protein driving the pathogenesis. The need for such combination therapy is likely to extend to other disease-driving proteins.

## Materials and methods

### Experimental procedures

#### Mouse models and preparation of brain extracts

All mouse procedures were approved by the Institutional Animal Care and Use Committee for Baylor College of Medicine and Affiliates. $Atxn1^{154Q/+}$,$Atxn1^{78Q/+}$ and $Atxn1^{-/-}$ mice have been previously described (*Lorenzetti et al., 2000*; *Watase et al., 2002*) and were backcrossed to C57BL/6 for more than ten generations. Mouse cerebella were dissected and lysed in 0.5% Triton buffer (0.5% Triton X-100, 50 mM Tris pH 8, 75 mM NaCl) supplemented with protease and phosphatase inhibitors (Sigma, St-Louis, Mo). The protein lysate was then incubated on ice for 20 min and centrifuged at 13,200 r.p.m. for 10 min at 4°C, and the supernatants were portioned into aliquots, snap-frozen, and stored at -80°C until used.

#### Rotarod assay

Motor coordination was assessed on the Rotarod assay as previously described (*Park et al., 2013*), with four trials a day (separated by 1 hr each) for 4 days. The tester was blinded to animal genotype and treatment.

## Immunotherapy

We used F11G3 and a control mouse IgM as antibodies for immunotherapy. Antibodies were administered at 5 mg/kg via intraperitoneal (i.p) injection once a week for six weeks. One week after completion of the treatment 12 mice per group were tested on the rotarod assay and sacrificed immediately afterward so that brains could be collected for biochemical and histopathological analysis. For survival studies, 12 mice per group were vaccinated once a week (5 mg/Kg) throughout their lifespan.

## Brain sections immunofluorescence

Paraffin sections were deparaffinized, rehydrated, and washed in 0.01 M PBS 3 times for 5 min each time. After blocking in normal goat serum for 1 hr, sections were incubated overnight with rabbit anti-ATXN1 antibody 11750 (1:700). The next day, the sections were washed in PBS 3 times for 10 min each and then incubated with goat anti-rabbit IgG Alexa Fluor 568 (1:700; Invitrogen) for 1 hr. The sections were then washed 3 times for 10 min each time in PBS before incubation overnight with mouse anti-oligomers F11G3 (1:300). The next day, the sections were washed in PBS 3 times for 10 min each before incubation with goat anti-IgM Alexa Fluor 488 (1:700; Invitrogen) for 1 hr. Sections were washed and mounted in Vectashield mounting medium with DAPI (Vector Laboratories). The sections were examined using a Zeiss LSM 710 confocal microscope.

## Immunohistochemistry

IHC was performed on paraffin-embedded sections. In brief, sections (5 μm) were deparaffinized and rehydrated. Primary antibodies were detected with biotinylated goat anti-mouse IgG (1:2000; Jackson ImmunoResearch Laboratories), biotinylated goat anti-mouse IgM (1:1500), or biotinylated goat anti-rabbit IgG (1:1800) (all from Jackson ImmunoResearch Laboratories) and visualized using an ABC reagent kit (Vector Laboratories, Burlingame, CA), according to the manufacturer's recommendations. Bright-field images were acquired using a Carl Zeiss Axio Imager M2 microscope, equipped with an Axio Cam MRc5 color camera (Carl Zeiss, Oberkochen, Germany). Sections were counterstained with hematoxylin (Vector Laboratories) for nuclear staining. The following antibodies were used for immunostaining: rabbit anti-oligomer antibody A-11 (1:600), mouse anti-oligomer antibody F11G3 (1:100), and mouse anti-calbindin antibody (1:450).

## Stereotaxic surgery

$Atxn1^{78Q/+}$, wild type or null Atxn1 mice were anaesthetized with a mixture of ketamine (110 mg/kg body weight) and xylazine (20 mg/kg body weight) in saline. Bilateral stereotaxic injections of 2.5 μl brain extract (3.9 μg/μL) from $Atxn1^{154Q/+}$or Wild Type mice were placed with a Hamilton syringe into the cerebellum (From Bregma; Post -6.0 mm, lat +/- 2.0 mm, dv -2.2 mm). Injection speed was 1.25 μl/minute and the needle was kept in place for an additional 2 min before it was slowly withdrawn. The surgical area was cleaned with sterile saline, the incision was sutured, and the mice were monitored until recovery from anesthesia. If not otherwise stated, 8 mice/group were used. Injected material was obtained from mouse brain extracts prepared from the cerebellum of aged 30-week-old $Atxn1^{154Q/+}$ and age-matched, non-transgenic wild-type control mice. All animal experiments were in compliance with protocols approved by the local Animal Care and Use Committee.

## Cell-based seeding assay

A stable Daoy mRFP-ATXN1(82Q) cell line was generated as previously described (*Park et al., 2013*). Cells were plated in 24-well plates (2*104 cells/ml). 24 hr later, cells were treated with 2.5 μl brain extract (3.9 μg/μL) from $Atxn1^{154Q/+}$or Wild Type mice. After 10 hr of treatment, cells were fixed with methanol at −80°C for 45 min. RFP-positive inclusions ranging from 350 to 900 nm were considered oligomeric; inclusions larger than 900 nm and not detected by F11G3 were considered fibrillar. One hundred cells were quantified per group in triplicates. Analyses were manually performed with Image J. For blocking assays, each sample was mixed with F11G3 antibody (2.5 mg/ml) in a ratio 1:1 (vol/vol) for 1 hr and then added to the cells.

## ELISA

For ELISA, plates were coated with 10 µl of the soluble fraction of brains using 0.1 M sodium bicarbonate (pH 9.6) as a coating buffer, followed by incubation for 1 hr at 37°C, washing three times with Tris-buffered saline with very low (0.01%) Tween 20 (TBS-T), and then blocking for 1 hr at 37°C with 10% BSA. The plates were then washed three times with TBS-T; F11G3 (1:500), A-11 (1:1000), 11750 (1:2000) or Tubulin (1:2000) antibodies (diluted in 5% nonfat milk in TBS-T) were added and allowed to react for 1 hr at 37°C. The plates were then washed three times with TBS-T, and 100 µl of horseradish peroxidase-conjugated anti-mouse IgM, anti-mouse IgG or anti-rabbit IgG (diluted 1:10,000 in 5% nonfat milk in TBS-T; Promega, Madison, WI) were added, followed by incubation for 1 hr at 37°C. Finally, plates were washed 3 times with TBS-T and developed with 3,3,5,5-tetramethylbenzidine (TMB-1 component substrate) from KPL (Gaithersburg, MD). The reaction was stopped with 100 µl of 1 M HCl, and samples were read at 450 nm.

## Analysis of ATXN1 pathology

To determined the percent of PCs with ATXN1 oligomers, 5 µm brain sections were stained with anti-Calbindin antibody as described above to quantify the total number of PCs in the cerebellum. The number of PCs positive for Calbindin was consider as the 100%. The adjacent sections were immunostained for oligomers using F11G3 or A-11. For the quantification of nuclear inclusions, we stained ATXN1 using 11750 antibody and performed nuclear staining with hematoxylin. We considered the total amount of nuclei in the cortex as the 100% of nucleus. Data were analyzed using *post-hoc* test.

## Statistical analysis

Experimental analysis and data collection were performed in a blinded fashion. p-values were determined using the appropriate statistical method via GraphPad Prism, as described throughout the manuscript. For simple comparisons, Student's t-test was used. For multiple comparisons, ANOVA followed by the appropriate *post hoc* analysis were utilized. All data is presented as mean ± SEM. *, ** and *** denote $p < 0.05$, $p < 0.01$ and $p < 0.001$, respectively.

# Acknowledgements

We thank the members of the Zoghbi, Orr and Kayed laboratories for suggestions and discussions, and V Brandt for critical reading of the manuscript. This work was supported by a Howard Hughes Medical Institute Collaborative Innovation Awards grant, the Robert A and Renee E Belfer Family Foundation and grant NIH/NINDS R01 NS027699-17. The NIH/NINDS 3R01 NS027699-25S1 and 1K22NS092688-01 to CALR. MWCR gratefully acknowledges The Canadian Institutes of Health Research Fellowship (201210MFE-290072-173743). We also appreciate the assistance of the confocal microscopy and mouse behavioral cores of the Baylor College of Medicine (BCM) Intellectual and Developmental Disabilities Research Center (1U54 HD083092).

# Additional information

### Competing interests

HYZ: Senior editor, *eLife.* The other authors declare that no competing interests exist.

### Funding

| Funder | Grant reference number | Author |
| --- | --- | --- |
| Howard Hughes Medical Institute | | Huda Y Zoghbi |
| Robert and Renee E Belfer Family Foundation | | Cristian A Lasagna-Reeves Maxime WC Rousseaux Paymaan Jafar-Nejad |
| National Institute of Neurological Disorders and Stroke | R01 NS027699-17 | Jeehye Park Huda Y Zoghbi |

The funders had no role in study design, data collection and interpretation, or the decision to submit the work for publication.

## Author contributions

CAL-R, HYZ, Conception and design, Acquisition of data, Analysis and interpretation of data, Drafting or revising the article; MWCR, MJG-M, LV-V, JP, LS, PJ-N, RR, Performed experiments, Acquisition of data, Analysis and interpretation of data, Drafting or revising the article; HTO, RK, Assisted in design and interpretation of experiments, Acquisition of data, Analysis and interpretation of data, Drafting or revising the article

## Ethics

Animal experimentation: This study was performed in strict accordance with the recommendations in the Guide for the Care and Use of Laboratory Animals of the National Institutes of Health. All of the animals were handled according to approved Institutional Animal Care and Use Committee (IACUC) protocols (#AN-1013) of Baylor College of Medicine

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
