## [Decision Letter]

Thank you for submitting your work entitled "Ataxin-1 oligomers induce local spread of pathology and decreasing them by passive immunization slows SCA1 phenotypes" for peer review at *eLife*. Your submission has been favorably evaluated by a Senior editor, a Reviewing editor, and three reviewers.

The reviewers have discussed the reviews with one another and the Reviewing editor has drafted this decision to help you prepare a revised submission.

Summary:

This is a timely and intriguing follow up to the author's recent paper showing that oligomers are primary drivers of toxicity. Here the authors test whether injected oligomers can drive regional propagation of SCA1 pathology and whether passive immunotherapy can reverse disease phenotypes. The authors show data that oligomers can propagate within a localized region and that this propagation is limited only to neighboring areas. Immunotherapy results provide some indication that this approach could have limited efficacy. The major points are that for ataxin-1, propagation does not appear to be the major driver of toxicity, therefore immunotherapy against extracellular oligomers would require additional therapeutic strategies in combination. This is a very important point relating to treatment that passive immunotherapy may not be curative because neurons will form toxic oligomeric entities internally and that one would need to target abnormal accumulation of polyQ ATXN1.

Overall some type of more direct test of either the injected oligomers being the driver of seeding or whether immunotherapy impacts oligomers directly versus reducing mutant ATXN1 levels would have been helpful. We would welcome any additional experimental back up to address the major criticisms mentioned below but consider it crucial that the authors indicate very clearly the limitations of the work in their final manuscript by addressing in their revised text the questions raised in the essential revisions section.

Essential revisions:

1) Cerebellar extracts from control or Atxn1 154Q/+ mice were injected into Atxn1 78Q/+ or wild type mice and limited spreading to neighboring areas within the cerebellum and brain stem was detected. The authors state that the fact ATXN1 levels were the same indicates that oligomers were induced by the seeding effect of ATXN1-154Q soluble oligomers. It would have been helpful to use similar approaches as used in the initial published studies to deplete lysates of ataxin-1 (using oligomer antibody) to show that the effect was directly a consequence of oligomers, but minimally the previous study and discussion of this depletion experiment should be discussed. There are other factors that could provide this effect including inflammatory cytokines or other external (and limiting) factors. Further, it is not mentioned until the Discussion whether this localized spreading had an effect on behavior or neurodegeneration in the Atxn1 78Q/+ mice and should be discussed in the Results.

2) The authors perform passive immunotherapy with F11G3 using IP injection in 154Q/+ mice. Decreased oligomers and ATXN1 were observed in cerebella. What is status at 4 weeks when injected? The authors postulate that the antibody targets extracellular oligomeric entities. This experiment could also be done in the 78Q/+ mice using the conditions with injected lysates. This would provide a direct test of whether immunotherapy targeted the oligomers.

3) Could the phenotypic effect of the passive immunity experiment be due to the decrease of ATXN1 (panel 2D), not the oligomers since increasing the oligomers in the 78Q background does not result in a phenotype?

4) It is briefly mentioned that the increase in oligomers in the 78Q background post-injection is not associated with neurodegeneration or motor incoordination. The authors interpret this to mean that the "toxic entity" (presumably the 154Q) needs to be continuously expressed to observe a phenotype. As the 78Q expansion is abnormal yet insufficient to cause disease in the mouse, presumably due to their short lifespan (subsection “ATXN1 oligomers seed the formation of new endogenous ATXN1 oligomers in vivo”, first paragraph), seeding the formation of oligomers was insufficient to increase the pathogenicity of this abnormal repeat as hypothesized by the authors. This would call into question the true pathogenicity of the oligomers if their presence does not alter phenotype. Can the authors comment on whether pure 78Q oligomers are likely formed following seeding from the 154Q lysate or whether all the oligomers likely contain one or more 154Q molecules? If pure 78Q oligomers are formed post-seeding, then why is there no phenotype? If the 78Q is simply insufficient to cause disease, does injection of the oligomers back into 154Q itself worsen that phenotype?

---

## [Author Response]

*Essential revisions:*

1) Cerebellar extracts from control or Atxn1 154Q/+ mice were injected into Atxn1 78Q/+ or wild type mice and limited spreading to neighboring areas within the cerebellum and brain stem was detected. The authors state that the fact ATXN1 levels were the same indicates that oligomers were induced by the seeding effect of ATXN1-154Q soluble oligomers. It would have been helpful to use similar approaches as used in the initial published studies to deplete lysates of ataxin-1 (using oligomer antibody) to show that the effect was directly a consequence of oligomers, but minimally the previous study and discussion of this depletion experiment should be discussed. There are other factors that could provide this effect including inflammatory cytokines or other external (and limiting) factors. Further, it is not mentioned until the Discussion whether this localized spreading had an effect on behavior or neurodegeneration in the Atxn1 78Q/+ mice and should be discussed in the Results.

In our previous study, we pre-incubated F11G3 with *Atxn1^154Q/+^*brain homogenate to demonstrate that the seeding effect was a direct consequence of ATXN1 oligomers in the brain (please refer to Figure 3 in (Lasagna-Reeves et al., 2015)). In the current study, using this same strategy, we performed a cell-based seeding assay using the *Atxn1^154Q/+^*brain homogenate (injected material) alone or pre-incubated with F11G3 antibody. Pre-incubating the *Atxn1^154Q/+^*brain homogenate with F11G3 completely abolished the seeding nature of the homogenate (Figure 1—figure supplement 1; subsection “ATXN1 oligomers seed the formation of new endogenous ATXN1 oligomers in vivo”, second paragraph). This suggests that the seeding/propagation observed in vivo is due to the oligomers present in the injected material. Nevertheless, in the manuscript we acknowledge that other factors in the injected material such as inflammatory responses could also play a role in propagation. We also discussed the necessity to perform future studies where lysate depleted of ATXN1 oligomers are injected in *Atxn1^78Q/+^*mice.

With regards to whether the localized spreading had an effect on behavior in *Atxn^78Q/+^*mice, we performed Rotarod analysis to test for the presence of motor coordination defects in these mice (Figure 1—figure supplement 2). We found that – consistent with our pathological findings – only mild and unsustained defects were observed in the *Atxn^78Q/+^* mice injected with *Atxn1^154Q/+^*brain homogenate (in the last paragraph of the aforementioned subsection).

*2) The authors perform passive immunotherapy with F11G3 using IP injection in 154Q/+ mice. Decreased oligomers and ATXN1 were observed in cerebella. What is status at 4 weeks when injected? The authors postulate that the antibody targets extracellular oligomeric entities. This experiment could also be done in the 78Q/+ mice using the conditions with injected lysates. This would provide a direct test of whether immunotherapy targeted the oligomers.*

Our new data demonstrate that ATXN1 oligomers are already present in *Atxn1^154Q/+^*mice cerebellum at 4 weeks of age. Nevertheless the amount of oligomers at this age is modest in comparison with *Atxn1^154Q/+^*mice at 11 weeks of age (Figure 2—figure supplement 1; subsection “Passive immunotherapy decreases ATXN1 oligomer pathology and improves motor coordination”, first paragraph).

We agree with the reviewers’ observation regarding the administration of F11G3 in *Atxn^78Q/+^*mice after lysate injection in order to provide direct test of whether immunotherapy targeted oligomers. However, this experiment would take several months as we would have to restart essentially all experiments (to have the proper littermate and age matched controls). Thus, to answer this question as a proxy, we showed that immunotherapy using F11G3 in *Atxn1^154Q/+^*mice specifically targets ATXN1 oligomers but not monomeric ATXN1 (Figure 3; in the aforementioned paragraph). These data lend further support that the effects observed throughout these experiments are from the immunotherapy specifically targeting ATXN1 oligomers.

*3) Could the phenotypic effect of the passive immunity experiment be due to the decrease of ATXN1 (panel 2D), not the oligomers since increasing the oligomers in the 78Q background does not result in a phenotype?*

Our new data demonstrate how the immunotherapy in *Atxn1^154Q/+^*mice specifically targets ATXN1 oligomers but not monomeric ATXN1 (Figure 3; subsection “Passive immunotherapy decreases ATXN1 oligomer pathology and improves motor coordination”, first paragraph). Therefore, the decrease of total ATXN1 levels measured by ELISA (Figure 2) is mainly due to a decrease in the levels of ATXN1 oligomers.

Regarding the fact that no phenotype is observed in the *Atxn1^78Q/+^*mice injected with *Atxn1^154Q/+^*brain homogenate; we added new comments in the Results section (subsection “ATXN1 oligomers seed the formation of new endogenous ATXN1 oligomers in vivo”, last paragraph). Furthermore, we address this point in more detail in question 4.

*4) It is briefly mentioned that the increase in oligomers in the 78Q background post-injection is not associated with neurodegeneration or motor incoordination. The authors interpret this to mean that the "toxic entity" (presumably the 154Q) needs to be continuously expressed to observe a phenotype. As the 78Q expansion is abnormal yet insufficient to cause disease in the mouse, presumably due to their short lifespan (subsection “ATXN1 oligomers seed the formation of new endogenous ATXN1 oligomers in vivo”, first paragraph), seeding the formation of oligomers was insufficient to increase the pathogenicity of this abnormal repeat as hypothesized by the authors. This would call into question the true pathogenicity of the oligomers if their presence does not alter phenotype. Can the authors comment on whether pure 78Q oligomers are likely formed following seeding from the 154Q lysate or whether all the oligomers likely contain one or more 154Q molecules? If pure 78Q oligomers are formed post-seeding, then why is there no phenotype? If the 78Q is simply insufficient to cause disease, does injection of the oligomers back into 154Q itself worsen that phenotype?*

At this time, we cannot ascertain the exact nature of the oligomeric entities in the brains of injected animals without proper genetic tools (such as having differentially tagged knockin mice for *Atxn1^154Q/+^* and *Atxn1^78Q/+^* and probing for their respective tags biochemically). Nevertheless, we comment in the results section whether pure 78Q ATXN1 oligomers are likely formed following seeding from the *Atxn1^154Q/+^*brain lysate or whether all the oligomers likely contain one or more 154Q ATXN1 molecules (subsection “ATXN1 oligomers seed the formation of new endogenous ATXN1 oligomers in vivo”, last paragraph). In the same section, we also discussed that the newly formed oligomers are insufficient to cause disease, possibly either because the healthy neuronal environment in *Atxn1^78Q/+^* mice can counteract the toxic effect of these oligomers or because they are structurally different from 154Q toxic oligomers.

We have not had to opportunity to inject oligomers into *Atxn1^154Q/+^*mice yet given their sensitivity to stereotaxic surgery. However, considering the abundance of published studies from several proteinopathies indicating that seeding inoculate into presymptomatic mice accelerates neurodegeneration (Guo and Lee, 2014), we believe that the injected material would accelerate and perhaps worsen phenotypes in *Atxn1^154Q/+^*mice.